# IMPROVING ROBUSTNESS AND ACCURACY WITH RETROSPECTIVE ONLINE ADVERSARIAL DISTILLATION

## ABSTRACT

Adversarial distillation (AD), transferring knowledge of a robust teacher model to a student model, has emerged as an advanced approach to improving robustness against adversarial attacks. However, AD in general suffers from the high computational complexity of pre-training the robust teacher as well as the inherent trade-off between robustness and natural accuracy (*i.e.*, accuracy on clean data). To address these issues, we propose retrospective online adversarial distillation (ROAD). ROAD exploits the student itself of the last epoch and a natural model (*i.e.*, a model trained with clean data) as teachers, instead of a pre-trained robust teacher in the conventional AD. We revealed both theoretically and empirically that knowledge distillation from the student of the last epoch allows to penalize overly confident predictions on adversarial examples, leading to improved robustness and generalization. Also, the student and the natural model are trained together in a collaborative manner, which enables to improve natural accuracy of the student more effectively. We demonstrate by extensive experiments that ROAD achieved outstanding performance in both robustness and natural accuracy with substantially reduced training time and computation cost.

## 1 INTRODUCTION

Deep neural networks (DNNs) have achieved great success in various applications such as computer vision (He et al., 2016; Goodfellow et al., 2014), natural language processing (Sutskever et al., 2014; Vaswani et al., 2017), and reinforcement learning (Mnih et al., 2013; Chen et al., 2021). However, Szegedy et al. (2013) showed that DNNs are vulnerable to adversarial attacks (Goodfellow et al., 2015; Dong et al., 2018; Carlini & Wagner, 2017; Madry et al., 2018), which are small perturbations added to natural inputs to deceive the models and cause incorrect predictions consequently. These attacks are significant threats especially in high-stakes contexts including autonomous driving (Sitawarin et al., 2018) and financial systems (Fursov et al., 2021).

Adversarial training (AT) has served as an effective solution to defend against the adversarial attacks (Madry et al., 2018; Gowal et al., 2020; Pang et al., 2021). It improves robustness of DNNs by training them with adversarial examples crafted from themselves. To further enhance their robustness, even for compact models, adversarial distillation (AD) has attracted increasing attention recently (Goldblum et al., 2020; Zhu et al., 2021; Zi et al., 2021; Maroto et al., 2022; Huang et al., 2023). Analogous to knowledge distillation (KD) (Hinton et al., 2015), AD adopts the teacher-student framework, in which the teacher model is pre-trained via AT and provides additional supervision to the student model for improving its robustness. Surprisingly, even when the teacher and student have the same capacity, AD enhances the robustness beyond the student trained alone. This suggests that AD not only compresses a high capacity model into a compact one but also enables to achieve extra robustness.

However, AD has a fatal drawback: it requires a lot of training time and computing resources. Most AD methods follow the two-stage training strategy, *i.e.*, pre-training a robust teacher through AT, and then transferring knowledge of the teacher to a student. Hence, AD typically demands at least twice as much training time as AT. This drawback make AD impractical for applications with limited computing resources or tight deployment schedules. Also, although AD enhances the robustness of the student through insights from the teacher, it is still limited in resolving the inherent trade-off between robustness and natural accuracy (*i.e.*, accuracy on clean data).

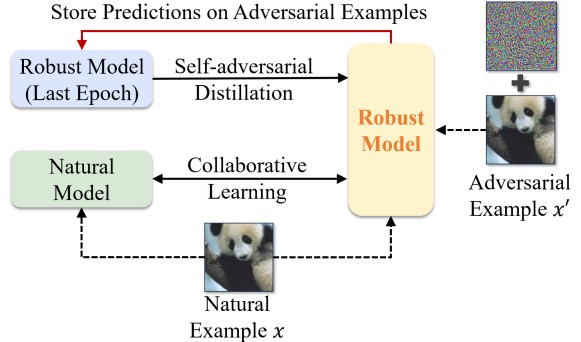

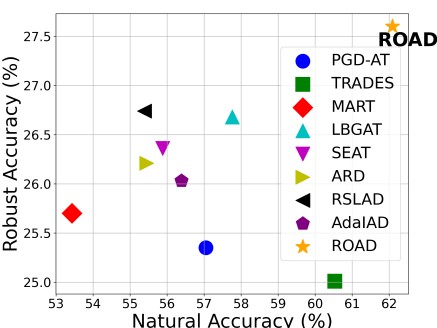

Figure 1: An overview of ROAD. The student makes predictions on both adversarial and natural examples. Then, for training, it is guided by two teachers: the student itself of the last epoch and a natural model trained collaboratively with the student. ROAD thus demands no pre-trained robust teacher.

Figure 2: Robustness versus natural accuracy of ResNet-18 on CIFAR-100. The robust accuracy is measured by AutoAttack (Croce & Hein, 2020). ROAD outperformed widely used AT and AD methods in both robustness and natural accuracy.

To address these limitations, we propose a new AD method coined retrospective online adversarial distillation (ROAD). Unlike the conventional AD using a pre-trained robust teacher, ROAD trains a robust model using knowledge distilled from two teachers: the model itself of the last epoch and an additional natural model, *i.e.*, a standard model trained with clean data, as illustrated in Figure 1. To be specific, the robust model is trained using soft labels generated by linear interpolation between its predictions in the past epoch and true one-hot labels. Through theoretical and empirical analysis, we find that this simple method penalizes overly confident predictions on adversarial examples, thereby enhancing its generalization ability and robustness. Moreover, we employ a collaborative learning strategy to train the robust model and the natural model simultaneously. This enables the natural model to be aware of the robust model and consequently provide more friendly knowledge to the robust model. Note that these two teachers are substantially cheaper than the teacher pre-trained via AT in the conventional AD. Thanks to the use of the two teachers, ROAD achieved outstanding performance in both robustness and natural accuracy (Figure 2) with substantially reduced training time and computation cost (Figure 5(c) and 5(d)). Our major contribution is three-fold:

- We propose ROAD, a new single-stage AD method based on retrospective self-distillation and collaborative learning, to address the chronic issues of the conventional AD approach.
- ROAD demonstrated superior performance in both robustness and natural accuracy with diverse network architectures on two datasets and three different adversarial attacks.
- ROAD allows to substantially reduce overall training time and computation cost of AD. To be specific, it requires about half the training time and memory of the previous best AD method.

## 2   RELATED WORK

**Adversarial Training.** Adversarial training has proven to be an effective defense against adversarial attacks. One fundamental approach is PGD adversarial training (Madry et al., 2018), using the Projected Gradient Descent algorithm. Subsequent advancements have introduced regularization terms to enhance performance. For instance, Zhang et al. (2019) achieved a principled trade-off between robustness and accuracy, while Wang et al. (2020) focused on improving robustness by revisiting misclassified examples. Kannan et al. (2018) improved robustness by a technique called adversarial logit pairing. Other approaches involve additional unlabeled data utilization (Carmon et al., 2019; Uesato et al., 2019; Gowal et al., 2021; Wang et al., 2023) or perturbing the weight of the model (Wu et al., 2020) or utilize extra models (Chen et al., 2020; Cui et al., 2021; Arani et al., 2020; Rade & Moosavi-Dezfooli, 2022; Dong et al., 2022; Wang & Wang, 2022). However, AT methods cannot ensure high robustness for small-sized models. Regarding this, ROAD demonstrates distinctiveness by showing high robustness not only in large models but also in small models.

**Adversarial Distillation.** The goal of adversarial distillation is to train a small-sized student model to mimic both the natural and robust predictions of a larger-sized robust teacher model. The initial works is Goldblum et al. (2020) which propose Adversarially Robust Distillation (ARD) to achieve robustness by comparing the model's robust prediction with teacher's natural prediction. Zi et al. (2021) compared conventional AT methods from a distillation perspective, emphasizing the advantages of using soft labels to achieve higher robustness. Based on this observation, they proposed Robust Soft Label Adversarial Distillation (RSLAD), which involves training the student model using soft labels generated from the teacher's predictions. In addition, Zhu et al. (2021) pointed out that a robust teacher might provide unreliable predictions for adversarial examples crafted by the student model and proposed Introspective Adversarial Distillation (IAD), in which the teacher's predictions are partially trusted. Liu et al. (2022) proposed Mutual Adversarial Training (MAT), which trains multiple robust models collaboratively to share the knowledge achieved from each adversarial examples. Lastly, Huang et al. (2023) proposed Adaptive Adversarial Distillation (AdaAD), which adaptively searches for inner maximization results by comparing the differences in predictions of student and teacher models. AD methods can be an attractive alternatives to enhance the robustness of end devices. However, the inherent two-stage process and associated computational inefficiency still conveys an inappropriate impression.

## 3 RETROSPECTIVE ONLINE ADVERSARIAL DISTILLATION

ROAD consists of two components: retrospective self-adversarial distillation using the robust model itself of the last epoch to improve robustness, and collaborative learning with a natural model to recover natural accuracy. We first elaborate on each of the two components in Section 3.1 and Section 3.2, respectively, and then describe the overall training objective for ROAD in Section 3.3.

### 3.1 SELF-ADVERSARIAL DISTILLATION FROM LAST EPOCH

AD has been acknowledged as an effective way to achieving extra robustness by improving generalization ability. However, pre-training the robust teacher model through AT demands an extremely large amount of training time. For instance, pre-training a 10-step PGD model requires roughly 11 times more forward-backward passes compared with the natural training. Additionally, loading both the teacher and student during the distillation process significantly increases GPU memory usage.

To address these challenges, we introduce a simple yet efficient approach to improving robustness, *self-adversarial distillation from the last epoch*. Our distillation scheme does not require a teacher model as the student becomes its own teacher. Instead, it leverages the predictions on adversarial examples made by the robust model (*i.e.*, the student) itself in the past. This approach eliminates the necessity of training an additional robust model. Specifically, it mixes the past predictions for adversarial examples with their one-hot labels by interpolation ratio $\lambda$. Ideally $\lambda$ should increase gradually as the predictions from previous epochs becomes more accurate. Considering that, we adopt a monotonically increasing schedule based on the sine function for $\lambda$. Then, the soft labels for robust model at the $t$-th epoch are given by

$$\tilde{y}_t = (1 - \lambda_t)y + \lambda_t p_{t-1}^{\text{rob}}(x'_{t-1}), \tag{1}$$

where $p_{t-1}^{\text{rob}}(x'_{t-1})$ is the output of the robust model for the adversarial example $x'_{t-1}$ at $(t-1)$-th epoch. The model trains with these soft labels instead of conventional one-hot labels.

### 3.1.1 THEORETICAL ANALYSIS

We carefully analyze the role of the adversarial predictions at the last epoch as supervision. To this end, we first discuss the relationship between over-confidence and robustness in AT. Although robust models are less over-confident than natural models in prediction (Grabinski et al., 2022), their predictions still tend to be overly confident as they are trained with one-hot labels. Stutz et al. (2020) pointed out that AT *overfits* to *experienced* norm bounded adversarial examples (*e.g.*, $\ell_\infty$ norm bounded adversarial examples) and performs poorly on other $\ell_p$ norm bounded adversarial examples or those crafted with a larger perturbation bound. Chen et al. (2020) claimed that the cause of robust overfitting, as discussed by Rice et al. (2020), is the model's tendency to overfit to adversarial examples during the early stages of the training process, resulting in lack of generalizability. Therefore, it can be inferred that the over-confidence acts as a factor that diminishes the

generalization capability of robust models and thus hampers the gain of robustness. We claim that our method resolves this problem by penalizing updating the model when its prediction confidence is boosted drastically for input adversarial examples during training. This phenomenon can be explained by gradient rescaling factor, following propositions presented by Tang et al. (2020) and Kim et al. (2021). The gradient rescaling factor is defined as the ratio of the $\ell_1$ norm of the loss gradient with respect to the logit during AD to that during training with one-hot labels.

**Proposition 1.** Given a $K$-class classification problem, let $p'_{t,i}$ be the output of the robust model for an adversarial example of class $i$ ($i = 1, 2, \ldots, K$) at the $t$-th epoch, and $GT$ be the ground truth class. The gradient rescaling factor is then derived as

$$\frac{\sum_i \left| \partial_i^{AD,t} \right|}{\sum_i |\partial_i|} = 1 - \lambda_t \left( \frac{1 - p'_{t-1,GT}}{1 - p'_{t,GT}} \right) \equiv 1 - \lambda_t \left( \frac{\gamma_{t-1}}{\gamma_t} \right),$$

where $\partial_i$ and $\partial_i^{AD,t}$ represent the gradients of a logit of class $i$ when trained with standard one-hot labels and the proposed soft labels at epoch $t$, respectively. Also, $\gamma$ indicates the inverse confidence of the prediction for the ground truth class. The detailed derivation is in Appendix B.1. Note that $\frac{\gamma_{t-1}}{\gamma_t}$ becomes larger as the prediction confidence on the adversarial example has increased significantly compared to the last epoch. This refers that our method assigns relatively smaller weights to examples that exhibit substantial improvement. Consequently, our method acts as a countermeasure, preventing the model's predictions from becoming overly confident and thus possess superior calibration performance, which has been known as an attribute of robust models (Grabinski et al., 2022; Wu et al., 2023).

### 3.1.2 EMPIRICAL ANALYSIS

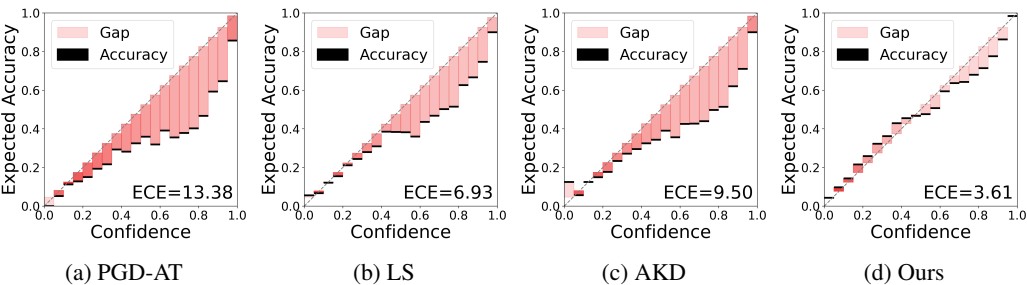

Figure 3: Reliability diagrams for PGD-AT, LS, AKD, and our method on CIFAR-100. We note the ECE (lower is better) on the right bottom side of the diagram.

To empirically verify the results of our theoretical analysis, we prepare four ResNet-18 models trained with PGD-AT (Baseline), PGD-AT with label smoothing (LS), adversarial knowledge distillation (AKD), and self-adversarial distillation from the last epoch (Ours). LS (Szegedy et al., 2016) is known to mitigate the over-confidence problem by imposing uncertainty on one-hot labels. AKD (Maroto et al., 2022) is an AD method that trains a model using as supervision combinations of adversarial predictions from a pre-trained robust model and one-hot labels. We compare our technique with these prior arts in terms of calibration performance since these arts aim to suppress over-confidence by adopting soft labels which is similar with ours. An ideally well calibrated model would provide high confidence in correct classifications, and low confidence in wrong classifications. To evaluate the calibration performance, we use the expected calibration error (ECE) (Naeini et al., 2015) as our metrics. Let $M$ and $n$ denote the number of confidence interval bins and that of the individual samples, respectively, where each bin contains samples whose confidences fall within the corresponding interval $\left[ \frac{m-1}{M}, \frac{m}{M} \right]$. Then, ECE is defined as

$$\text{ECE} = \sum_{m=1}^{M} \frac{|B_m|}{n} \left| \text{Acc}(B_m) - \text{Conf}(B_m) \right|.$$

Predictions on adversarial examples are collected along with their respective confidence levels. Further experiment details are in the Appendix B.2. The reliability diagrams (Guo et al., 2017) of the

four methods and their ECE scores are presented in Figure 3; the smaller the gap in the reliability diagram, the better the calibration of the model. As shown in the figure, our method demonstrates significantly low over-confidence in its predictions on adversarial examples compared to other methods. Furthermore, we observe that our method achieved a notably lower ECE score in comparison, indicating superior calibration performance.

## 3.2 COLLABORATIVE LEARNING WITH NATURAL MODEL

The trade-off between robustness and natural accuracy has been a longstanding issue of AT (Tsipras et al., 2019; Zhang et al., 2019; Yang et al., 2020). This issue persists in AD as it adopts the robust teacher model trained using AT. It is hard to expect that a teacher with high robustness can also be a proper supervisor that helps improve natural accuracy. Thus, to achieve both robustness and natural accuracy through AD, it is reasonable to consider for a student to benefit from the guidance of two teachers: one trained naturally and the other trained through AT. A few prior studies (Chen et al., 2020; Zhao et al., 2022) share this concept to mitigate the trade-off using a pre-trained natural model as the frozen natural model. However, experimental results presented in the literature (Zi et al., 2021; Maroto et al., 2022) demonstrate that distilling knowledge from the static natural model can reduce robustness, indicating that it is not the proper approach.

In this paper, we take a different view towards the robust and natural models, and adopt the framework of online distillation (Zhang et al., 2018; Guo et al., 2020; Cui et al., 2021). Instead of using the pre-trained and frozen natural model as a teacher, we treat the natural model as a peer so that both models exchange their knowledge during the training process. We focus on the fact that a model trained by AT and that trained differ in training schemes and data, leading to distinct knowledge representations. By employing mutual learning between these models, we expect that the robust model can acquire knowledge from the natural model without compromising its robustness.

Meanwhile, exchanged knowledge takes on differing roles from each model's perspective. The natural model views the robust model's insights as a form of regularization. On the other hand, the robust model views the knowledge from the natural model as an alternative to one-hot labels. This asymmetry necessitates a careful balance in the quantity of knowledge exchanged during the collaborative training. Also, since a robust model is in general substantially poor in natural accuracy at early stages of training, knowledge transfer from the robust model to the natural counterpart may hinder their collaborative learning. We thus control the impact of the collaborative learning dynamically through a weight parameter $\lambda$ following the monotonically increasing schedule based on the sine function as in Section 3.1, where the weight is used to create soft labels based on predictions of the robust model. These soft labels are then utilized for training the natural model. This strategy mitigates the cold start issue and ensures effective knowledge transfer between the two models throughout the collaborative learning process. The soft labels for the natural model are given by

$$\hat{y}_t = (1 - \lambda_t)y + \lambda_t p_t^{\mathrm{rob}}(x_t), \tag{2}$$

where $p_t^{\mathrm{rob}}(x_t)$ is the output of the robust model for the natural example at the $t$-th epoch. We train the natural model using these soft labels $\hat{y}_t$, instead of one-hot labels, through the cross-entropy loss. While the natural model is trained using the soft labels, the robust model receives supervision from the natural model through a standard KL-divergence loss.

## 3.3 THE OVERALL OBJECTIVE

Incorporating the techniques that we suggested in the previous sections, the final objective function for ROAD is given by

$$\min_{\theta_{\mathrm{rob}}} \underbrace{\mathrm{CE}(f_{\theta_{\mathrm{rob}}}(x'), \tilde{y})}_{\text{Self-Guidance}} + \beta \cdot \underbrace{\mathrm{KL}(f_{\theta_{\mathrm{rob}}}(x'), f_{\theta_{\mathrm{rob}}}(x))}_{\text{Robustness Enhancement}} + \gamma \cdot \underbrace{\mathrm{KL}(f_{\theta_{\mathrm{rob}}}(x), f_{\theta_{\mathrm{nat}}}(x))}_{\text{Natural Model Guidance}}, \tag{3}$$

where $f_{\theta_{\mathrm{rob}}}$ is the robust model, $f_{\theta_{\mathrm{nat}}}$ is the natural model, hyper-parameter $\beta$ controls the trade-off between robustness and natural accuracy, and hyper-parameter $\gamma$ controls the amount of guidance. The training objective of ROAD contains three components: The first component is derived from Section 3.1, forcing the model to be less over-confident on adversarial examples and improving its generalization consequently. The second term is adopted to further improve robustness by minimizing the output distribution difference between adversarial examples and natural examples; for

---

**Algorithm 1** Retrospective Online Adversarial Distillation (ROAD)

---

**Require:** Robust model $f_{\theta_{\text{rob}}}$, Natural model $f_{\theta_{\text{nat}}}$, training dataset $D$, learning rate $\tau$, number of epochs $T$, batch size $m$, number of batches $M$, maximum perturbation bound $\epsilon$, attack iterations $K$, step size $\eta$, robust factor $\beta$, guidance factor $\gamma$.

1: **for** epoch $= 1, \ldots, T$ **do**
2:      **for** mini-batch $= 1, \ldots, M$ **do**
3:          Sample a mini-batch $\{(x_i, y_i)\}_{i=1}^m$ from $D$
4:          **for** $i = 1, \ldots, m$ **do**
5:              $x_i' \leftarrow x_i + \epsilon, \quad \epsilon \sim \text{Uniform}(-\epsilon, \epsilon)$
6:              **for** $k = 1, 2, \ldots, K$ **do**
7:                  $x_i' \leftarrow \Pi_{B_\epsilon(x_i)} \left( x_i' + \eta \cdot \text{sign} \left( \nabla_{x_i'} \text{CE}(f_{\theta_{\text{rob}}}(x_i'), y_i) \right) \right)$
8:              **end for**
9:          **end for**
10:          Obtain robust soft labels $\tilde{y}$ using Eq. (1).
11:          Obtain natural soft labels $\hat{y}$ using Eq. (2).
12:          $\theta_{\text{rob}} \leftarrow \theta_{\text{rob}} - \tau \nabla_{\theta_{\text{rob}}} (\text{CE}(f_{\theta_{\text{rob}}}(x'), \tilde{y}) + \beta \cdot \text{KL}(f_{\theta_{\text{rob}}}(x'), f_{\theta_{\text{rob}}}(x))$
                                      $+ \gamma \cdot \text{KL}(f_{\theta_{\text{rob}}}(x), f_{\theta_{\text{nat}}}(x).\text{detach}())$
13:          $\theta_{\text{nat}} \leftarrow \theta_{\text{nat}} - \tau \nabla_{\theta_{\text{nat}}} (\text{CE}(f_{\theta_{\text{nat}}}(x), \hat{y})$
14:          Save predictions of $f_{\theta_{\text{rob}}}$ on $x'$
15:      **end for**
16: **end for**

---

this purpose, we utilized the KL divergence loss, following prior studies (Zhang et al., 2019; Wang et al., 2020). This regularization term causes loss in natural accuracy as a trade-off for improved robustness. Nevertheless, this loss of accuracy can be recovered by the subsequent term. Finally, the last component enhances natural accuracy by matching the output distributions of the robust model and its peer natural model. To this end, we adopt KL-divergence to effectively distill the knowledge from the natural model. The complete algorithm is described in Algorithm 1.

## 4 EXPERIMENTAL RESULTS

### 4.1 SETUP

We mainly use ResNet-18 (He et al., 2016) and MobileNetV2 (Sandler et al., 2018) architectures and train the models with the SGD optimizer with momentum of 0.9. The batch size is set to 128.

ROAD is compared with five AT methods, PGD-AT (Madry et al., 2018), TRADES (Zhang et al., 2019), MART (Wang et al., 2020), LBGAT (Cui et al., 2021), and SEAT (Wang & Wang, 2022), and five AD methods, ARD (Goldblum et al., 2020), KD+SWA (Chen et al., 2020), IAD (Zhu et al., 2021), RSLAD (Zi et al., 2021), and AdaIAD (Huang et al., 2023), as well as the combination of AdaAD and IAD (Zhu et al., 2021).

We conduct our evaluations on CIFAR-100 and CIFAR-10 (Krizhevsky et al., 2009). For PGD-AT, TRADES, MART, and ROAD, we set the number of epochs to 120 with weight decay 3.5e-3 and the learning rate starts from 0.01 and is divided by 10 at the 75, 90, and 100 epochs. We clarify that the PGD-AT model is trained with different settings with the one mentioned in Section 3.1, focusing on evaluating robustness in this section while assessing generalization ability in Section 3.1. The robust factor $\beta$ is set to 6.0 for TRADES and MART. For LBGAT and SEAT, we directly comply with the official implementation. For ROAD, we fix $\beta$ to 6.0 and set $\gamma$ to 3.0 and 5.0 for CIFAR-100 and CIFAR-10, respectively. $\lambda_t$ follows sine increasing schedule starting from 0 to 0.8. Meanwhile, for the AD methods, we set the number of epochs to 200 with weight decay 5e-4. The learning rate starts from 0.1 and is divided by 10 at the 100, 150, and 175 epochs except KD+SWA. We directly use the PGD-AT model as the teacher model. For KD+SWA, we additionally use the NAT model as the natural teacher model. As recommended in Goldblum et al. (2020), we set the hyper-parameter $\alpha$ of ARD and AdaIAD to 1.0 and distillation temperature $\tau$ to 5.0 and 30.0 for CIFAR-100 and CIFAR-10, respectively. In other training details, we strictly follow the settings from the original papers. For natural model, we train with natural images and the learning rate starts from 0.1 and is divided by 10 at the 75, 90, and 100 epochs.

## 4.2 PERFORMANCE WITH COMPACT ARCHITECTURES

We first evaluate the robustness of our method in compact size architectures, ResNet-18 and MobileNetV2. We report the results on CIFAR-100 and CIFAR-10 in Table 1 and Table 2, respectively. Regardless of the architectures or datasets, our ROAD demonstrates the best performance in most cases. Performance results in AA indicate that ROAD is robust under not only in white-box attacks but also in black-box attacks. The AD methods show improvement in terms of robustness compared to the PGD-AT teacher, but they exhibit a decrease in natural accuracy. In contrast, ROAD improves the natural accuracy by **1.56%** and **0.51%** on ResNet-18 and MobileNetV2 respectively, compared to other AT or AD methods. This demonstrates the significant role of collaborative learning with the natural model in mitigating the trade-off between robustness and natural accuracy.

Table 1: Validation results of ResNet-18 and MobileNetV2 models on CIFAR-100 trained with different methods. The best and second performances are marked in **bold** and underlined respectively.

| Model | Method | NAT | PGD-20 | PGD-100 | MIM-10 | AA |
|-------|--------|-----|--------|---------|--------|-----|
| RN-18 | NAT | 77.10 | 0.0 | 0.0 | 0.01 | 0.0 |
| | PGD-AT | 57.05 | 30.27 | 30.22 | 31.16 | 25.35 |
| | TRADES | 60.53 | 29.96 | 29.87 | 30.65 | 25.01 |
| | MART | 53.43 | 31.86 | 31.74 | 32.31 | 25.70 |
| | LBGAT | 57.76 | 33.11 | 33.03 | 33.51 | 26.68 |
| | SEAT | 55.88 | 31.33 | 31.33 | 31.82 | 26.36 |
| | ARD | 55.45 | 31.01 | 30.92 | 31.82 | 26.21 |
| | KD+SWA | 58.94 | 30.42 | 30.36 | 31.17 | 26.76 |
| | IAD | 54.59 | 31.45 | 31.47 | 32.17 | 26.51 |
| | RSLAD | 55.39 | 31.63 | 31.52 | 32.28 | 26.74 |
| | AdaIAD | 56.39 | 30.85 | 30.80 | 31.63 | 26.03 |
| | **ROAD** | **62.09** | **33.73** | **33.81** | **34.43** | **27.60** |
| MN-V2 | NAT | 75.96 | 0.0 | 0.0 | 0.09 | 0.0 |
| | PGD-AT | 56.26 | 29.18 | 29.08 | 30.27 | 24.40 |
| | TRADES | 59.06 | 29.44 | 29.32 | 30.05 | 24.29 |
| | MART | 48.50 | 30.66 | 30.61 | 30.83 | 23.94 |
| | LBGAT | 53.40 | 29.34 | 29.27 | 29.68 | 23.32 |
| | SEAT | 54.60 | 30.61 | 30.61 | 31.12 | 25.43 |
| | ARD | 51.14 | 27.77 | 27.65 | 28.52 | 23.16 |
| | KD+SWA | 54.73 | 28.78 | 28.72 | 29.50 | 24.62 |
| | IAD | 49.58 | 27.68 | 27.59 | 28.30 | 22.66 |
| | RSLAD | 53.07 | 30.84 | 30.75 | 31.68 | 25.84 |
| | AdaIAD | 55.12 | 29.86 | 29.65 | 30.56 | 24.76 |
| | **ROAD** | **59.57** | **32.44** | **32.27** | **33.02** | **25.98** |

Table 2: Validation results of ResNet-18 and MobileNetV2 models on CIFAR-10 trained with different methods. The best and second performances are marked in **bold** and underlined respectively.

| Model | Method | NAT | PGD-20 | PGD-100 | MIM-10 | AA |
|-------|--------|-----|--------|---------|--------|-----|
| RN-18 | NAT | 94.73 | 0.0 | 0.0 | 0.01 | 0.0 |
| | PGD-AT | 83.63 | 51.92 | 51.72 | 53.60 | 48.76 |
| | TRADES | 82.77 | 53.83 | 53.61 | 55.27 | 49.77 |
| | MART | 80.42 | 54.89 | **54.62** | 56.15 | 48.72 |
| | LBGAT | 78.11 | 54.26 | 54.08 | 55.37 | 49.92 |
| | SEAT | 83.49 | 54.40 | 54.44 | 55.92 | 50.78 |
| | ARD | 82.76 | 51.58 | 51.30 | 53.33 | 48.81 |
| | KD+SWA | 84.14 | 52.77 | 52.47 | 54.66 | 49.91 |
| | IAD | 82.05 | 53.82 | 53.68 | 55.12 | 49.77 |
| | RSLAD | 83.13 | 53.64 | 53.26 | 55.58 | 50.61 |
| | AdaIAD | 83.11 | 52.34 | 51.94 | 53.92 | 49.15 |
| | **ROAD** | **84.42** | **54.93** | 54.56 | **56.43** | **50.91** |
| MN-V2 | NAT | 93.06 | 0.0 | 0.0 | 0.0 | 0.0 |
| | PGD-AT | 82.57 | 50.45 | 50.17 | 52.20 | 47.34 |
| | TRADES | 81.17 | 52.05 | 51.95 | 53.36 | 48.64 |
| | MART | 77.48 | 53.34 | 53.28 | 54.34 | 46.87 |
| | LBGAT | 72.63 | 49.78 | 49.74 | 50.49 | 46.11 |
| | SEAT | 81.70 | 52.73 | 52.54 | 54.19 | 49.16 |
| | ARD | 79.46 | 48.23 | 47.94 | 49.95 | 45.33 |
| | KD+SWA | 81.44 | 51.52 | 51.33 | 53.26 | 48.51 |
| | IAD | 79.02 | 49.96 | 49.82 | 51.29 | 46.10 |
| | RSLAD | 81.93 | 51.81 | 51.62 | 53.53 | 48.81 |
| | AdaIAD | 81.87 | 51.06 | 50.90 | 52.60 | 47.91 |
| | **ROAD** | **82.77** | **53.72** | **53.45** | **54.91** | **49.27** |

### 4.3 PERFORMANCE WITH HIGHER CAPACITY ARCHITECTURE

In this section, we extend our evaluation on higher capacity architecture, WRN-28-10 (Zagoruyko & Komodakis, 2016). For PGD-AT, TRADES, MART, and ROAD, we adjust the weight decay to 5e-4 and the learning rate starts from 0.1. Other training details remain the same as described in Section 4. We report the results on CIFAR-100 in Table 3. A similar trend is shown as in the previous experiments. ROAD shows superiority on natural accuracy ranging from a minimum of **2.27%** to a maximum of **8.53%** compared to other models. Furthermore, it exhibits the best robustness even in white-box attacks and ranks second in AA. This result confirms that our method consistently demonstrates superior performance in both robustness and natural accuracy across different architectures. In Appendix D, additional experimental results demonstrate the superior performance of our method.

Table 3: Validation results of WRN-28-10 models on CIFAR-100 trained with different methods. The best and second performances are marked in **bold** and underlined respectively.

| Method | NAT | PGD-20 | PGD-100 | MIM-10 | AA |
|--------|-----|--------|---------|--------|-----|
| PGD-AT | 61.36 | 31.37 | 31.22 | 32.55 | 27.63 |
| TRADES | 60.10 | 32.23 | 32.17 | 32.89 | 27.60 |
| MART | 55.16 | 33.65 | 33.55 | 33.98 | 28.19 |
| LBGAT | 59.96 | 34.84 | 34.84 | 35.27 | 28.87 |
| SEAT | 59.72 | 34.46 | 34.41 | 34.97 | 29.52 |
| ARD | 60.01 | 32.60 | 32.43 | 33.75 | 28.88 |
| RSLAD | 59.78 | 33.90 | 33.74 | 34.93 | **29.91** |
| AdaIAD | 61.42 | 32.43 | 32.31 | 33.50 | 28.58 |
| **ROAD** | **63.69** | **35.10** | **35.06** | **35.97** | 29.66 |

### 4.4 ABLATION STUDIES

In this section, we study the importance of each component in ROAD through several experiments.

**Effect of different scheduling strategies.** We study the effect of epoch-wise interpolation ratio $\lambda$ scheduling. In addition to the sine increasing strategy used in our method, we prepare two simple strategies. One is fixing $\lambda$ at the final value, and the other is linearly increasing it. As shown in Figure 4(a), the sine increasing strategy shows the best robustness. Since the sine increasing strategy reaches the final value more quickly than the linear increasing strategy, therefore, it greatly benefits from the effects of self-distillation starting from the midpoint of the training process. In contrast, the fixed strategy exhibits the lowest performance in both natural accuracy and robustness, indicating that the cold start issue could actually hinder learning.

**Effect of transferring asymmetric knowledge.** Next, we also study the effect of asymmetric knowledge transfer between the natural and robust model in ROAD. To verify its effectiveness, we prepare the symmetric version of ROAD: the natural model achieves knowledge via not soft labels but KL-divergence, typically seen in conventional online distillation. We reuse $\gamma$ for simplicity and symmetric knowledge transfer. As shown in Figure 4(b), ROAD significantly outperforms the symmetric version of ROAD in natural accuracy regardless of the value of $\gamma$.

**Impact of soft labels.** We prepare three variants of ROAD: (1) where we removed the first soft labels $\tilde{y}$ to exclude self-distillation from predictions of the last epoch, (2) where we removed the second soft labels $\hat{y}$ to prevent natural model achieve knowledge from robust model through collaborative learning and (3) where we removed both $\tilde{y}$ and $\hat{y}$, replacing them with one-hot labels. As demonstrated in Figure 4(c), both variants show lower robustness than ROAD. This suggests that self-distillation enables the model to enhance robustness. Furthermore, it can be inferred that when the natural model unilaterally conveys knowledge to the robust model, although it may be helpful for natural accuracy, it causes a detrimental effect on robustness.

**Impact of hyper-parameter $\gamma$.** Here, we conduct an experiment to analyze the impact of hyper-parameter $\gamma$. While fixing $\beta$ to 6.0, we vary $\gamma$ from 1.0 to 6.0. The results are demonstrated in Figure 5(a) and Figure 5(b). It is noteworthy that natural accuracy consistently increases as the value of $\gamma$ increases. Furthermore, ROAD achieves the best robust accuracy with $\gamma = \{3.0, 5.0\}$ on CIFAR-100 and CIFAR-10, respectively.

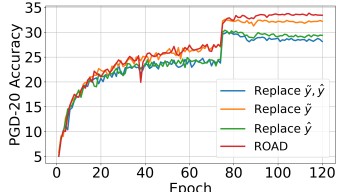

| Method | NAT | AA |
|---|---|---|
| ROAD (Fixed) | 59.66 | 26.07 |
| ROAD (Linear) | 62.56 | 27.23 |
| ROAD (Sine) | 62.09 | 27.60 |

| Method | NAT | AA |
|---|---|---|
| ROAD ($\gamma = 1$)(KL) | 57.91 | 27.30 |
| ROAD ($\gamma = 2$)(KL) | 60.00 | 27.65 |
| ROAD ($\gamma = 3$)(KL) | 60.45 | 27.36 |
| ROAD ($\gamma = 4$)(KL) | 60.65 | 27.36 |
| ROAD | 62.09 | 27.60 |

(a) Effect of scheduling interpolation ratio $\lambda$.    (b) Effect of transferring robust knowledge to natural model    (c) Effect of soft labels compared with one-hot labels

Figure 4: Comprehensive ablation study of each components of ROAD on CIFAR-100 with ResNet-18. We verify our methods

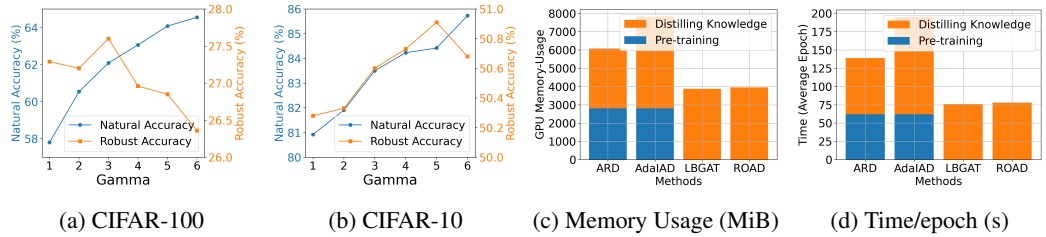

(a) CIFAR-100    (b) CIFAR-10    (c) Memory Usage (MiB)    (d) Time/epoch (s)

Figure 5: Comprehensive experiment results of ROAD. (a) and (b) are experiment results of impact of hyper-parameter $\gamma$ in ROAD on CIFAR-100 and CIFAR-10 with ResNet-18, respectively. (c) and (d) are the gpu memory usage and training time cost of ARD, AdaIAD, LBGAT, and ROAD with ResNet-18 on CIFAR-100.

## 4.5 EVALUATION ON COMPUTATIONAL COMPLEXITY

In this section, we compare the computational costs of ROAD with two adversarial distillation methods, ARD and AdaIAD, and one online distillation method, LBGAT. We conduct experiments on a single NVIDIA 3090 GPU and maintain consistent implementation details as described in Section 4.1, excluding the evaluation process. For ARD and AdaIAD, we include the cost of pre-training the teacher model. The results are presented in Figure 5(c) and Figure 5(d). From the results, we observe that the computational cost of ROAD is relatively lower than that of ARD and AdaIAD. This is because ROAD does not require a pre-training process although it trains low cost natural model simultaneously. Furthermore, even when excluding the pre-training process, AdaIAD still consumes more time and memory as it requires multiple forward-backward passes of teacher model to craft adversarial examples. Meanwhile LBGAT exhibits a slightly lower computational cost and time consumption, the difference is negligible considering the superior performance of ROAD. Therefore, we can conclude that ROAD is more suitable to resource-constrained environments.

## 5 CONCLUSION

In this paper, we address the drawbacks of most existing adversarial distillation methods. We point out that conventional adversarial distillation methods require enormous computational cost to pre-train a robust teacher model. Furthermore, student models trained with these methods also suffer from the inherent trade-off between robustness and natural accuracy. Based on this discussion, we propose Retrospective Online Adversarial Distillation (ROAD), a novel self-adversarial distillation method to train a robust model which has high performance on natural accuracy. ROAD attempts to get guidance from the predictions on adversarial examples of last epoch and collaboratively trained natural model to improve robustness and natural accuracy, respectively. Extensive experiments reveal that ROAD exhibits outstanding performance on both natural accuracy and robustness compared with both AT and AD methods regardless of the dataset or size of the architecture.

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

## A    RELATED WORK (CONTINUED)

**Self-Knowledge Distillation.** In general, KD has traditionally relied on a large and high-performing teacher model. To overcome this limitation, several papers proposed self-knowledge distillation, where models achieves dark knowledge from themselves. Yuan et al. (2020) argued that KD is a variant of label smoothing regularization and introduces Teacher-free Knowledge Distillation. Kim et al. (2021) suggested that past epoch predictions could be used as soft labels instead and proposed Progressive Self-Knowledge Distillation. Shen et al. (2022) shared a similar concept with previous works, leveraging the predictions of the last mini-batch as soft labels to obtain more up-to-date information during the training process.

## B    MORE DETAILS OF USING PREDICTIONS OF THE LAST EPOCH

### B.1    PROOF OF PROPOSITION 1.

**Proposition 1.** (*Restated*) Given a $K$-class classification problem, let $p'_{t,i}$ be the output of the robust model for an adversarial example of class $i$ ($i = 1, 2, \ldots, K$) at the $t$-th epoch, and $GT$ be the ground truth class. The gradient rescaling factor is then derived as

$$\frac{\sum_i \left|\partial_i^{AD,t}\right|}{\sum_i |\partial_i|} = 1 - \lambda_t \left(\frac{1 - p'_{t-1,GT}}{1 - p'_{t,GT}}\right) \equiv 1 - \lambda_t \left(\frac{\gamma_{t-1}}{\gamma_t}\right),$$

where $\partial_i$ and $\partial_i^{AD,t}$ represent the gradients of a logit of class $i$ when trained with standard one-hot labels and the proposed soft labels at epoch $t$, respectively. Also, $\gamma$ indicates the inverse confidence of the prediction for the ground truth class.

*Proof.* First we note that for the logit $z'_{t,i}$ is the prediction on adversarial example corresponding to a specific class $i$ at $t$-th epoch. When training with typical one-hot label, the gradient of the logit is

$$\frac{\partial \mathcal{L}}{\partial z'_{t,i}} = p'_{t,i} - y_i, \tag{4}$$

where $\mathcal{L}$ is the cross-entropy loss, $p'_{t,i}$ is the predicted probability for class $i$, and $y_i$ is the true label. If we mix the last epoch prediction on adversarial example with one-hot label, the gradient for the AD loss for class $i$ at $t$-th epoch is given by

$$\frac{\partial \mathcal{L}^{AD}}{\partial z'_{t,i}} = \partial_{t,i}^{AD} = (1 - \lambda_t)(p'_{t,i} - y_i) + \lambda_t(p'_{t,i} - p'_{t-1,i}), \tag{5}$$

where $\lambda_t$ is a factor which is less than 1. The gradient for the target class GT is given by

$$\partial_{t,GT}^{AD} = (1 - \lambda_t)(p'_{t,GT} - 1) + \lambda_t(p'_{t,GT} - p'_{t-1,GT}) = (p'_{t,GT} - 1) - \lambda_t(p'_{t,GT} - 1). \tag{6}$$

In addition, the gradient for non-target classes is given by

$$\partial_{t,i}^{AD} = (1 - \lambda_t)(p'_{t,i} - 0) + \lambda_t(p'_{t,i} - p'_{t-1,i}) = p'_{t,i} - \lambda_t p'_{t-1,i}. \tag{7}$$

If we set $\lambda_t$ such that Equation 7 is non-negative, then $\lambda_t$ can be expressed as

$$\lambda_t \leq \min_i \left(\frac{p'_{t,i}}{p'_{t-1,i}}\right) \leq \frac{p'_{t,GT} - 1}{p'_{t-1,GT} - 1}. \tag{8}$$

The sum of $\ell_1$ norm of the AD gradient for the entire class can be expressed as

$$\sum_i \left|\partial_{t,i}^{AD}\right| = 2(1 - p'_{t,GT}) - 2\lambda_t(1 - p'_{t-1,GT}). \tag{9}$$

Regarding that sum of $\ell_1$ norm of the gradients trained with one-hot labels are $\sum_i |\partial_i| = 2(1 - p'_{t,GT})$, we can compute the gradient rescaling factor and can be expressed as

$$\frac{\sum_i \left|\partial_{t,i}^{AD}\right|}{\sum_i |\partial_i|} = 1 - \lambda_t \left(\frac{1 - p'_{t-1,GT}}{1 - p'_{t,GT}}\right) \equiv 1 - \lambda_t \left(\frac{\gamma_{t-1}}{\gamma_t}\right). \tag{10}$$

## B.2 Implementation Details of Section 3.1

We run ResNet-18 models for 120 epochs and train with SGD with momentum of 0.9. The batch size is set to 128 and weight decay is set to 5e-4. We conduct the experiments on both CIFAR-100 and CIFAR-10. The learning rate is set to 0.1 and is divided by 10 at the 75, 90, 100 epochs. We use a relatively low weight decay to ensure a fair comparison of the generalization abilities of each method. Adversarial examples are generated by using PGD-20. For label smoothing (LS), we use a smoothing factor of 0.1. For AKD, we select label mixing factor of 0.8 which is stated on Maroto et al. (2022). We use the PGD-AT model as the teacher model.

## B.3 Performance Evaluation on Calibration on CIFAR-10

To broadly evaluate the calibration performance of our method, we also conduct additional experiments on CIFAR-10. The reliability diagram and ECE are shown in Figure 6. While all models exhibit a lower generalization gap and ECE compared to CIFAR-100, we still observe over-confidence in AT and AKD. On the other hand, our method demonstrates outstanding calibration performance and the gap is hard to be visualized.

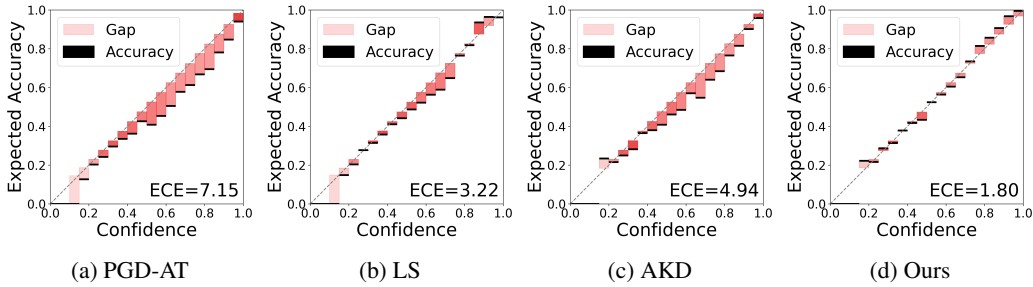

(a) PGD-AT        (b) LS        (c) AKD        (d) Ours

Figure 6: Reliability diagrams for PGD-AT, LS, AKD, and our method on CIFAR-10. We note the ECE (lower is better) on the right bottom side of the diagram.

## C Evaluation Metrics

We select three adversarial attack methods to evaluate the robustness of the models, including PGD (Madry et al., 2018), MIM (Dong et al., 2018), and AutoAttack (AA) (Croce & Hein, 2020). The maximum perturbation bound $\epsilon$ is set to 8/255 for all the evaluation methods under the $\ell_\infty$ norm. The attack codes are implemented based on the Torchattacks library (Kim, 2020). During training process, we evaluate the model with PGD-20 and select the model with the highest robustness.

## D Further Performance Evaluation

### D.1 Performance Evaluation on Higher Capacity Architecture on CIFAR-10

We extended our experiment using a higher-capacity architecture on CIFAR-10, as discussed in Sec 4.3. All other training specifications are consistent with those detailed in Section 4.3. The outcomes for CIFAR-10 are presented in Table 4. ROAD displays a superior performance in natural accuracy, with a minimum of **0.22%** and a maximum of **7.34%** compared to other methods. ROAD demonstrates comparable or superior defense performance on adversarial attacks compared to its counterparts.

### D.2 Comparing Adversarial Distillation Methods with a Large Size Teacher

In this section, we evaluate our method under unfair conditions. Since AD was initially designed as a framework to distill robust knowledge from a teacher model with high capacity, we use the PGD-AT model from Section 4.3 as the teacher model. The other values remain identical to those in the previous section. We present our results in Table 5. As seen in the results, AD methods

Table 4: Validation results of WRN-28-10 models on CIFAR-10 trained with different methods. The best and second performances are marked in **bold** and underlined respectively.

| Method | NAT | PGD-20 | PGD-100 | MIM-10 | AA |
|---|---|---|---|---|---|
| PGD-AT | 86.14 | 54.31 | 53.98 | 56.39 | 51.7 |
| TRADES | 84.91 | 55.66 | 55.33 | 57.13 | 52.38 |
| MART | 79.72 | 56.73 | 56.64 | 57.81 | 51.11 |
| LBGAT | 80.31 | 55.71 | 55.53 | 57.06 | 52.44 |
| SEAT | 86.84 | 57.26 | 56.96 | **59.29** | **54.16** |
| **ROAD** | **87.06** | **57.44** | **57.18** | 58.95 | 53.98 |

demonstrate an increasing trend in natural accuracy and robustness as the capacity of the teacher model grows. Specifically, AdaIAD receives the most benefits on utilizing a higher capacity teacher model. Moreover, our method significantly surpasses the natural accuracy and robustness of models trained with AD. This reconfirms our method's superior performance, even when compared against AD models benefiting from additional knowledge through stronger teacher models.

Table 5: Validation results of AD methods on CIFAR-100 with various teacher models. The best and second performances are marked in **bold** and underlined respectively.

| Method | Student | Teacher | NAT | PGD-20 | PGD-100 | MIM-10 | AA |
|---|---|---|---|---|---|---|---|
| ARD | ResNet-18 | ResNet-18 | 55.45 | 31.01 | 30.92 | 31.82 | 26.21 |
| | ResNet-18 | WRN-28-10 | 57.63 | 30.11 | 29.84 | 31.26 | 26.37 |
| RSLAD | ResNet-18 | ResNet-18 | 55.39 | 31.63 | 31.52 | 32.28 | 26.74 |
| | ResNet-18 | WRN-28-10 | 59.36 | 30.78 | 30.59 | 31.64 | 26.93 |
| AdaIAD | ResNet-18 | ResNet-18 | 56.39 | 30.85 | 30.80 | 31.63 | 26.03 |
| | ResNet-18 | WRN-28-10 | 60.64 | 31.00 | 30.87 | 32.01 | 27.49 |
| ROAD | ResNet-18 | ResNet-18 | **62.09** | **33.73** | **33.81** | **34.43** | **27.60** |

### D.3 Performance Evaluation of ROAD on Calibration on CIFAR-100

To assess calibration performance of ROAD, we prepare two AD methods: ARD and RSLAD. We ensure a fair comparison by aligning the training details of ROAD, including learning rate, weight decay, and number of epochs, with those of ARD and RSLAD. Figure 7 demonstrates that ROAD achieves superior calibration performance with the lowest ECE. Moreover, ROAD tends to make under-confident predictions in contrast to the over-confident predictions commonly associated with the other methods.

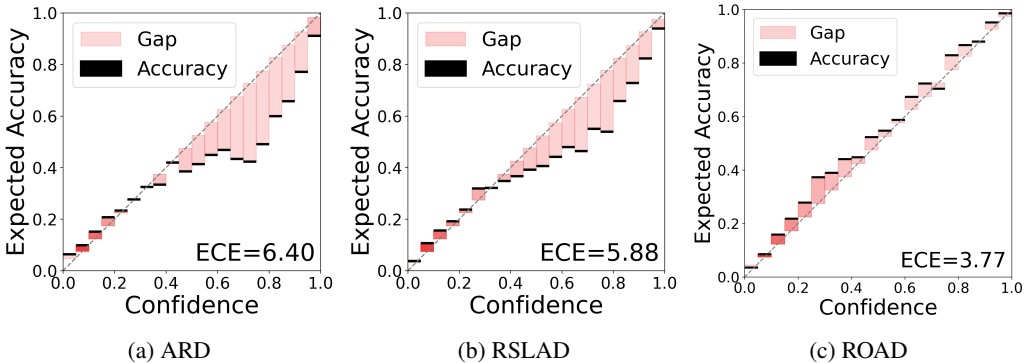

(a) ARD          (b) RSLAD          (c) ROAD

Figure 7: Reliability diagrams for ARD, RSLAD, and ROAD on CIFAR-100. We note the ECE (lower is better) on the right bottom side of the diagram.

# E ABLATION STUDIES (CONTINUED)

## E.1 EFFECTS ON UTILIZING LAST EPOCH PREDICTIONS.

Table 6 demonstrates the effectiveness of utilizing last epoch predictions of adversarial examples. We report the best performance achieved by the models, which are identical to the ones used in Section 3.1. While LS seems to contribute to enhancing robustness, its effect is marginal. In contrast, our method outperforms other methods significantly on natural accuracy and white-box attacks. Additionally, our method achieves comparable or better performance than the self-adversarial distillation method AKD in AA. This indicates that our method effectively leverages last epoch predictions, fulfilling the role of the teacher model.

Table 6: Ablation study with ResNet-18 student models on CIFAR-10 and CIFAR-100 with different methods. The best and second performances are marked in **bold** and underlined respectively.

| Dataset | Method | NAT | PGD-20 | PGD-100 | MIM-10 | AA |
|---|---|---|---|---|---|---|
| CIFAR-10 | PGD-AT | 82.63 | 51.54 | 51.25 | 53.13 | 47.98 |
| | LS | 82.44 | 52.20 | 52.01 | 53.90 | 48.75 |
| | AKD | 83.29 | 52.99 | 52.73 | 54.71 | 49.36 |
| | Ours | **83.93** | **54.00** | **53.69** | **55.75** | **49.95** |
| CIFAR-100 | PGD-AT | 57.23 | 28.84 | 28.72 | 29.56 | 25.13 |
| | LS | 57.10 | 29.32 | 29.24 | 30.34 | 25.17 |
| | AKD | **57.62** | 30.74 | 30.60 | 31.58 | **26.66** |
| | Ours | 57.39 | **31.45** | **31.42** | **32.02** | 26.17 |

## E.2 EFFECT OF DIFFERENT SCHEDULING STRATEGIES ON UTILIZING LAST EPOCH PREDICTIONS.

As shown in Table 7, utilizing last epoch predictions show an overall improvement in robustness compared to PGD-AT which trained with one-hot labels. This observation substantiates that utilizing last epoch predictions prevents the model to give overly confident predictions to adversarial examples and thus improve robustness. In addition, we observe that the fixed policy results in over 1% lower natural accuracy compared to other strategies. This observation can be attributed to the inaccuracy of predictions from initial epochs, which can impede learning. Meanwhile, the linear scheduling strategy exhibits slightly lower robustness compared to the sine-based strategy, suggesting the importance of the scheduling approach in balancing natural accuracy and robustness.

Table 7: Effect of scheduling interpolation ratio $\lambda$

| Method | NAT | AA |
|---|---|---|
| PGD-AT | 57.23 | 25.13 |
| Ours (Fixed) | 56.20 | 26.16 |
| Ours (Linear) | 57.55 | 25.88 |
| Ours (Sine) | 57.39 | 26.17 |

## E.3 IMPACT OF HYPER-PARAMETER $\beta$.

Here, we study the impact of the robustness factor $\beta$. We use the same training settings as Section 4.1. We fix $\gamma = 3.0$ and vary $\beta$ from 1.0 to 6.0. The results can be seen in Figure 8. As expected, as $\beta$ increases, the model achieves more robustness but also loses natural accuracy. However, even with low $\beta$ values, ROAD demonstrates competitive robustness compared to other methods with outstanding natural accuracy. For higher $\beta$ values, it exhibits both high robustness and high natural accuracy.

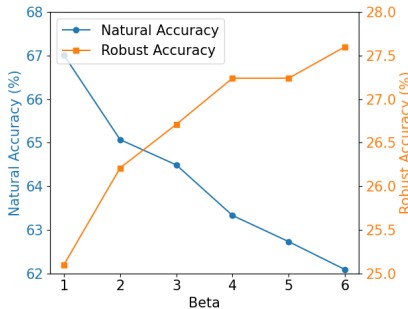

Figure 8: Sensitivity of robust factor $\beta$ on CIFAR-100 dataset.

## F STABILITY ACROSS MULTIPLE RUNS

In this section, we check the variance of ROAD. Table 8 provides a detailed comparison of two machine learning models, ROAD and TRADES, tested on the CIFAR-100 dataset. It showcases the results from five separate runs for each model, highlighting their performance stability and robustness. Results for ROAD and TRADES are presented side by side in each cell, allowing for direct comparison. The inclusion of average (Avg.) and standard deviation (Std.) for each method across the runs offers a summary of overall performance and variability, respectively. The results illustrate that ROAD not only maintains a steady performance across different methods but also outperforms TRADES in both natural accuracy and robustness.

Table 8: Validation results of ROAD (left) and TRADES (right) models across multiple runs on CIFAR-100.

|      |   | NAT | | PGD-20 | | PGD-100 | | MIM-10 | | AA | |
|------|---|-------|-------|-------|-------|-------|-------|-------|-------|-------|-------|
| Run  | 1 | 62.09 | 60.53 | 33.74 | 29.96 | 33.81 | 29.87 | 34.43 | 30.65 | 27.60 | 25.01 |
|      | 2 | 61.80 | 60.61 | 33.53 | 30.32 | 33.48 | 30.24 | 34.19 | 31.07 | 27.41 | 25.59 |
|      | 3 | 62.01 | 59.88 | 33.68 | 30.32 | 33.67 | 30.20 | 34.47 | 31.23 | 27.66 | 25.69 |
|      | 4 | 61.62 | 60.34 | 33.88 | 30.24 | 33.69 | 30.01 | 34.45 | 31.03 | 27.68 | 25.43 |
|      | 5 | 62.08 | 60.10 | 33.36 | 30.35 | 33.28 | 30.19 | 34.15 | 31.02 | 27.14 | 25.46 |
| Avg. |   | **61.92** | 60.29 | **33.63** | 30.24 | **33.58** | 30.10 | **34.33** | 31.00 | **27.49** | 25.44 |
| Std. |   | 0.07 | 0.23 | 0.18 | 0.15 | 0.18 | 0.10 | 0.13 | 0.20 | 0.20 | 0.24 |

