# OpenReview forum: "Improving Robustness and Accuracy with Retrospective Online Adversarial Distillation"
_ICLR.cc/2024/Conference — Submitted to ICLR 2024_

### Official Review · Reviewer_FC1d · 2023-10-30

**Soundness:** 3 good
**Presentation:** 3 good
**Contribution:** 2 fair
**Rating:** 5
**Confidence:** 5

**Summary:**

This paper studies the adversarial robustness via model distillation and proposes retrospective online adversarial distillation,  which exploits the student itself of the last epoch and a natural model as teachers to guide target model training. The paper proves the effectiveness of the proposed method through experiments and provides theoretical analysis.

**Strengths:**

1. The proposed method is single-stage adversarial distillation, but this is not the first time.
2. The authors provide a theoretical and experimental analysis of the label for robust overfitting.

**Weaknesses:**

1. The novelty of this paper is insufficient. It is highly overlapped with previous work, such as the use of soft labels, which was explored in the previous work [1]. Secondly, the use of online training methods was explored in the work of [2], and the use of natural models as teacher was explored in [3].

2. Lack of ablation experiments, such as the impact of natural models, and the selection of robust models (different checkpoints)

ref:
[1] Revisiting Adversarial Robustness Distillation: Robust Soft Labels
Make Student Better.
[2]. Alleviating Robust Overfitting of Adversarial Training With Consistency Regularization.
[3] Learnable Boundary Guided Adversarial Training.

**Questions:**

See weaknesses

---

> ### Author Response · Authors · 2023-11-16
> **Response to reviewer FC1d**
>
> We sincerely appreciate the reviewer for the thoughtful feedback.
>
> **The novelty of this paper is insufficient. It is highly overlapped with previous work, such as the use of soft labels, which was explored in the previous work [1]. Secondly, the use of online training methods was explored in the work of [2], and the use of natural models as teacher was explored in [3].**
>
> Thanks for your concerns regarding the novelty of our paper. However, ROAD distinguishes itself from prior works in two main aspects.
>
> Firstly, While there may be similarities with RSLAD in using soft labels, ROAD does not require an adversarially trained model to improve robustness, thereby offering a computational cost advantage. Thus, ROAD can be a promising technique to enhance robustness if we have no pre-trained teacher models. Also, considering that adversarial training typically takes several times longer than conventional training, it can be considered to possess sufficient novelty.
>
> Secondly, our approach diverges from methods that utilize a natural model by implementing an asymmetric knowledge transfer. This is based on the premise that usage of acquired knowledge to the natural and robust models are different as we mentioned in Section 3.2. Our experiments and ablation studies demonstrate that ROAD significantly outperforms other methods in terms of natural accuracy through this asymmetric knowledge transfer.
>
> **Lack of ablation experiments, such as the impact of natural models, and the selection of robust models (different checkpoints)**
>
> We appreciate your constructive feedback for experiment part. We have conducted several additional experiments including comprehensive ablation studies (effects of interpolation ratio scheduling, asymmetric knowledge transfer) and stability of ROAD. We have updated our results in our paper.
>
> Table 1 shows the effect of interpolation ratio scheduling. We have considered two more scheduling strategies (fixed, linear increasing)
>
> Table 1. Effect of interpolation ratio $\lambda$
> | Method  | NAT   | AA    |
> |---------|-------|-------|
> | Fixed   | 59.66 | 26.07 |
> | Linear  | 62.56 | 27.23 |
> | Sine    | 62.09 | 27.60 |
>
> Table 2 shows the effect of transferring asymmetric knowledge. We have prepared the symmetric version of ROAD: the natural model achieves knowledge via not soft labels but KL-divergence, typically seen in conventional online distillation. We have reused $\gamma$ for simplicity and symmetric knowledge transfer.
>
> Table 2. Effect of transferring robust knowledge to natural model
>
> | Method               | NAT   | AA    |
> |----------------------|-------|-------|
> | ROAD ($\gamma = 1$)(KL) | 57.91 | 27.30 |
> | ROAD ($\gamma = 2$)(KL) | 60.00 | 27.65 |
> | ROAD ($\gamma = 3$)(KL) | 60.45 | 27.36 |
> | ROAD ($\gamma = 4$)(KL) | 60.65 | 27.36 |
> | ROAD                 | 62.09 | 27.60 |
>
> Table 3 shows the stability of ROAD. We have trained TRADES and ROAD 5 times with different random seeds. As we can observe in Table 3, ROAD shows high stability while outperforms TRADES in both natural accuracy and robustness.
>
> Table 3. Variance across multiple reruns
>
> |  | Run | NAT (ROAD) | NAT (TRADES) | PGD-20 (ROAD) | PGD-20 (TRADES) | PGD-100 (ROAD) | PGD-100 (TRADES) | MIM-10 (ROAD) | MIM-10 (TRADES) | AA (ROAD) | AA (TRADES) |
> |--------|-----|------------|--------------|---------------|----------------|----------------|------------------|---------------|----------------|----------|-------------|
> |        | 1   | 62.09      | 60.53        | 33.74         | 29.96          | 33.81          | 29.87            | 34.43         | 30.65          | 27.60    | 25.01       |
> |        | 2   | 61.80      | 60.61        | 33.53         | 30.32          | 33.48          | 30.24            | 34.19         | 31.07          | 27.41    | 25.59       |
> |        | 3   | 62.01      | 59.88        | 33.68         | 30.32          | 33.67          | 30.20            | 34.47         | 31.23          | 27.66    | 25.69       |
> |        | 4   | 61.62      | 60.34        | 33.88         | 30.24          | 33.69          | 30.01            | 34.45         | 31.03          | 27.68    | 25.43       |
> |        | 5   | 62.08      | 60.10        | 33.36         | 30.35          | 33.28          | 30.19            | 34.15         | 31.02          | 27.14    | 25.46       |
> | Avg.   |     | **61.92**  | 60.29        | **33.63**     | 30.24          | **33.58**      | 30.10            | **34.33**     | 31.00          | **27.49**| 25.44       |
> | Std.   |     | 0.07       | 0.23         | 0.18          | 0.15           | 0.18           | 0.10             | 0.13          | 0.20           | 0.20     | 0.24        |

---

> > ### Comment · Reviewer_FC1d · 2023-11-22
> > **Reply to author**
> >
> > Thanks to the author for the reply. From the author's response and the original submitted manuscript, the proposed method combines some previous methods. the contribution is just incremental.
> > Another question is why adversarial distillation (especially using natural models as teachers) can significantly improve the natural accuracy of adversarial models. There are no theoretical explanations.
> >
> > So I keep my rating.

---

### Official Review · Reviewer_m9Uu · 2023-10-31

**Soundness:** 2 fair
**Presentation:** 3 good
**Contribution:** 2 fair
**Rating:** 3
**Confidence:** 5

**Summary:**

The paper presents a novel technique designed to address the challenges inherent in balancing the trade-off between robust accuracy and natural accuracy while considering the computational overhead associated with adversarial distillation methods. The proposed method, referred to as ROAD, introduces a single-step training approach that incorporates self-distillation, employing the previous epoch's network as the teacher model. Additionally, a natural model is concurrently trained to provide guidance in the context of natural images, thus enhancing the aforementioned trade-off. ROAD leverages the utilization of soft-labels to penalize overconfident predictions, fostering collaborative learning.

**Strengths:**

- The paper addresses an important topic in the adversarial training domain.
-  It is well written and easy to follow. The algorithms and methodology is well explained
- The results cover three different networks and attacks (including AA)

**Weaknesses:**

**Motivation:**
The assertion positing that overconfidence serves as a factor impeding the generalization capacity of robust models, as detailed in Section 3.1.1, warrants substantiation through references or empirical analysis within an AT context. It remains unclear how the resolution for penalizing overconfident predictions is incorporated via the utilization of soft labels.

It is worth noting that self-distillation has already demonstrated its efficacy in numerous studies pertaining to supervised and adversarial training methods. These approaches often involve the utilization of a prior model's time-stamp or an Exponential Moving Average (EMA) model for the purpose of regularization. Furthermore, online collaborative training has been well-established in the existing literature, e.g. ACT [2] and CAD [4] among others. The incorporation of soft labels and label-smoothing techniques in AT is studied, so it's somewhat perplexing, as their novelty is not readily apparent.

**Complexity:**

- The ROAD method still maintains two separate networks.
- It necessitates the retention of an additional copy of the network (from the last epoch) in GPU memory and, in aggregate, entails three supplementary forward propagation steps.

**Baselines:**

- The paper appears to lack references to several relevant baselines, encompassing both new and established methods [1-4].
In particular, the omission of comparative data with respect to collaborative and online distillation training techniques, such as ACT and MAT, is noteworthy. Moreover, the results of these techniques in the paper do not match those from the baseline references [1], specifically as presented in Table 3. The paper briefly mentions IAD but fails to provide a comparative analysis of its results.
- Further inconsistencies arise from comparisons with the RSLAD paper, wherein the AA and other attack-related metrics appear to surpass the corresponding figures reported in the ROAD paper. Similar discrepancies are apparent in the case of SEAT results.
- A comprehensive evaluation of the ROAD method would ideally encompass additional benchmarks, including but not limited to Weight Averaging [5] and AWP [6].

References:

[1] Mutual Adversarial Training: Learning together is better than going alone

[2] Adversarial Concurrent Training: Optimizing Robustness and Accuracy Trade-off of Deep Neural Networks

[3] RELIABLE ADVERSARIAL DISTILLATION WITH UNRELIABLE TEACHERS

[4] Improving adversarial robustness through a curriculum-guided reliable distillation

[5] ROBUST OVERFITTING MAY BE MITIGATED BY PROPERLY LEARNED SMOOTHENING

[6] Adversarial Weight Perturbation Helps Robust Generalization

**Questions:**

- The related works section requires an update, incorporating more relevant and recent works.
- Please review the baseline models and consider adding more baselines (check the previous section for reference).
- In Section 3.1.2, the RSLAD also utilizes soft labels. Can we access the reliability diagram for it and obtain further information about the differences between these two methods?
- In Section 3.3, the second term of the objective function resorts to TRADES (involving the KL loss between robust and natural accuracy). Is this necessary when a separate model exists solely for natural images?
- The hyperparameter lambda is based on a hypothesis (that robust models are substantially poor in natural accuracy at early stages of training, as discussed in Section 3.2). Can we see some supporting evidence or results for this hypothesis? Ablations with different schedules for lambda to evaluate its impact on the results would be beneficial.
- Is the primary improvement stemming from the natural model, self-distillation, or solely from soft-labels? Further ablation experiments are necessary by removing each component individually.
- In Figure 4, does the orange line include guidance from the natural model with one-hot labels? What are the results when both self and natural model guidance objectives are replaced with one-hot labels?
- In Section 4.5, the computational cost should be compared with other online distillation methods.
- In Figure 5 (a) and (b), is the observed effect due to gamma or lambda? Why is there a significant difference between the CIFAR-10 and CIFAR-100 datasets? Hyperparameters should ideally not be highly sensitive to datasets and settings to develop more generalizable solutions.
- clarification and explanation of the differences between various concepts related to label smoothing

---

> ### Author Response · Authors · 2023-11-16
> **Response to reviewer m9Uu part1**
>
> Thanks for your insightful comments and detailed review! We prepare pointwise answers for your concerns and questions.
>
> **It remains unclear how the resolution for penalizing overconfident predictions is incorporated via the utilization of soft labels.**
>
> We have provided theoretical proofs in the respective appendix and supported our claims empirically through reliability diagrams and ECE results. However, we acknowledge that this may not suffice for your assessment. Therefore, we have also included the calibration performance of ROAD to further demonstrate the effects of utilizing last epoch predictions in Appendix D.3.
>
> **Concerns of novelty.**
>
> Thanks for addressing your concerns about the novelty of our paper. Our method achieves novelty in two key aspects.
>  Firstly, unlike other approaches that require the addition of extra models to enhance robustness, such as those using EMA or prior model's timestamps, our method stands out by not necessitating the use of additional models. We only store the predictions of last epoch. This feature offers a clear advantage in terms of GPU memory usage efficiency and it is more applicable in devices with limited resources. Secondly, our approach diverges from conventional methods that employ a natural model and learns collaboratively by implementing asymmetric knowledge transfer. This is based on the premise that usage of acquired knowledge to the natural and robust models are different as we mentioned in Section 3.2. We conducted an ablation experiments of the effects and updated in our revised paper.
>
> **The ROAD method still maintains two separate networks. It necessitates the retention of an additional copy of the network (from the last epoch) in GPU memory and, in aggregate, entails three supplementary forward propagation steps.**
>
> Sorry about the ambiguity of our paper about the implementation details of ROAD. ROAD maintains only one separate network, the natural model. We store the past predictions of the last epoch instead of copying the network which can be shown in Figure 1. This can be also seen in our publicly provided code. Thus, as we can observe in the experiment section, ROAD does not require a substantial amount of memory. However, we can also consider using a separate model if the predictions are massive to store, such as tokens for text classification.
>
> **The paper appears to lack references to several relevant baselines, encompassing both new and established methods**
>
> Thanks for your constructive feedback. We agree with the importance and relevance of ACT and MAT and updated our related works in our revised paper.
>
> **Further inconsistencies arise from comparisons with the RSLAD paper, wherein the AA and other attack-related metrics appear to surpass the corresponding figures reported in the ROAD paper. Similar discrepancies are apparent in the case of SEAT results.**
>
> We would like to highlight that there are variations in the implementation details between the original paper and ours, such as the total number of epochs (reduced from 300 to 200), changes in learning rate scheduling (from divided by 10 at the 215th, 260th, and 285th epochs, to the 100th, 150th, and 175th epochs), and adjustments in weight decay (from 2e-4 to 5e-4) for a fair comparison. Also, please consider that the RSLAD paper reports similar results to our paper in Appendix Table 9 when RSLAD is trained through self-distillation (RN-18 --> RN-18). Furthermore, it should be noted that SEAT, in their official code, opted for a maximum perturbation bound of 0.031, as opposed to our choice of 8/255, and a step size of 0.007 instead of 2/255. These differences may significantly impact the performance of robustness.
>
> **A comprehensive evaluation of the ROAD method would ideally encompass additional benchmarks, including but not limited to Weight Averaging and AWP**
>
> Thanks for your recommendation ! Due to the lack of time and computational resources, we will conduct the experiments you suggested at a later date and update the paper accordingly.
>
> **Please review the baseline models and consider adding more baselines (check the previous section for reference).**
>
> Thanks for your nice suggestion. We add KD+SWA[1] and IAD[2] to our baseline which is similar with ROAD in the fact that using teacher models.
>
> **In Section 3.1.2, the RSLAD also utilizes soft labels. Can we access the reliability diagram for it and obtain further information about the differences between these two methods?**
>
> We have added the reliability diagrams of ARD, RSLAD, and ROAD in the appendix of our updated paper. The Expected Calibration Error (ECE) of ROAD is 3.77, demonstrating superior calibration performance with the lowest ECE. Furthermore, we observe that ROAD tends to generate under-confident predictions, in contrast to the over-confident predictions commonly associated with ARD and RSLAD.

---

> > ### Author Response · Authors · 2023-11-16
> > **Response to reviewer m9Uu part2**
> >
> > **In Section 3.3, the second term of the objective function resorts to TRADES (involving the KL loss between robust and natural accuracy). Is this necessary when a separate model exists solely for natural images?**
> >
> > Apologize for our misunderstandings. Could you please provide further clarification regarding your question about the definition of a 'separate model'?
> >
> > **The hyperparameter lambda is based on a hypothesis (that robust models are substantially poor in natural accuracy at early stages of training, as discussed in Section 3.2). Can we see some supporting evidence or results for this hypothesis? Ablations with different schedules for lambda to evaluate its impact on the results would be beneficial.**
> >
> > This is a nice suggestion. We have conducted additional ablation studies with two different schedules for lambda (constant, linear) and updated our paper. Here we provide the results below:
> >
> > | Method  | NAT   | AA    |
> > |---------|-------|-------|
> > | Fixed   | 59.66 | 26.07 |
> > | Linear  | 62.56 | 27.23 |
> > | Sine    | 62.09 | 27.60 |
> >
> > **Is the primary improvement stemming from the natural model, self-distillation, or solely from soft-labels? Further ablation experiments are necessary by removing each component individually.**
> >
> > We attribute the improvements primarily to the natural model and the self-distillation. By using self-distillation utilizing the predictions of last epoch, the robust model significantly enhances its calibration performance, thereby achieving additional robustness. Meanwhile, adopting the framework of online distillation, the robust model also achieves natural accuracy which is one of the drawbacks of adversarial training. While soft labels do contribute to improving robustness, the performance gap between ROAD and RSLAD, along with the distinct aspects observed in the reliability diagram, indicate that soft labels are not the main contributor to these improvements.
> >
> > **In Figure 4, does the orange line include guidance from the natural model with one-hot labels? What are the results when both self and natural model guidance objectives are replaced with one-hot labels?**
> >
> > Yes. Thanks for your positive question and we include the results when both self and natural model guidance objectives are replaced with one-hot labels in our updated paper.
> >
> > **In Section 4.5, the computational cost should be compared with other online distillation methods.**
> >
> > We also agree with your thoughts, and we have included the computational cost of LBGAT which also uses online distillation. LBGAT has slightly lower computational cost and time for training compared to ROAD, but it is marginal.
> >
> > **In Figure 5 (a) and (b), is the observed effect due to gamma or lambda? Why is there a significant difference between the CIFAR-10 and CIFAR-100 datasets? Hyperparameters should ideally not be highly sensitive to datasets and settings to develop more generalizable solutions.**
> >
> > It is due to gamma. We do not modify the values of lambda. We agree that ROAD can be sensitive to types of datasets. However, we believe that this problem can be mitigated by a naive hyper-parameter tuning.
> >
> > Ref
> >
> > [1] Robust Overfitting may be mitigated by properly learned smoothening, ICLR 2021
> > [2] Reliable Adversarial Distillation with Unreliable Teachers, ICLR 2022

---

> > > ### Comment · Reviewer_m9Uu · 2023-11-22
> > > **Response to the rebuttal**
> > >
> > > I appreciate the authors' diligent efforts in conducting additional experiments and presenting their findings in the rebuttal. The clarification provided regarding the separate network (where predictions are saved instead of the previous epoch-model) is noted and appreciated.
> > >
> > > However, I have some reservations regarding certain claims made in the rebuttal, particularly concerning the assertion that the method stands out by not necessitating additional models. My concern arises from the fact that there remains the utilization of an additional natural model along with additional prediction memory. In contrast, there exist alternative adversarial distillation (AD) methods that operate with only two networks, without the incorporation of extra memory, and some employ weight averaging techniques without the use of additional natural models.
> > >
> > > Furthermore, I observed discrepancies in the training of RSLAD, specifically with different hyperparameters than those originally specified in the paper. To ensure a fair and accurate comparison, it would be advisable to adhere to the hyperparameters reported as the best-tuned in the original paper, ensuring consistency in the perturbation bound.
> > >
> > > Given the overlapping fundamental concepts shared with ACT (and partially with MAT), it might be insightful to include a comparative analysis, considering that ACT involved robust and natural model distillation but without the incorporation of soft labels.
> > >
> > > Regarding the AWP results, I would like to inquire about the rationale behind retraining the model for comparison purposes instead of utilizing the results from the original work for a comparative analysis.
> > >
> > > The observed sensitivity to hyperparameters raises a significant concern. Considering the aforementioned points, the claims of novelty in the rebuttal still appear to be subject to question. Therefore, I tend to keep my initial rating.
> > >
> > > I hope these comments and suggestions are helpful in refining the manuscript further. I appreciate the authors' dedication and willingness to address these concerns.

---

> > > > ### Author Response · Authors · 2023-11-23
> > > > **Response to reviewer m9Uu**
> > > >
> > > > Thank you so much for your thoughtful feedback on our paper.
> > > > We truly appreciate your constructive comments and suggestions and are grateful for the effort you put into reviewing our work.

---

### Official Review · Reviewer_zap7 · 2023-11-01

**Soundness:** 3 good
**Presentation:** 3 good
**Contribution:** 3 good
**Rating:** 6
**Confidence:** 4

**Summary:**

The paper introduces a novel method called "Retrospective Online Adversarial Distillation" (ROAD) to improve the robustness of Deep Neural Networks (DNNs) against adversarial attacks while also maintaining high natural accuracy (accuracy on clean data). Unlike conventional Adversarial Distillation (AD) methods which involve training a robust teacher model and then transferring the knowledge to a student model, ROAD utilizes the student model from the last epoch and a natural model (trained with clean data) as teachers.

**Strengths:**

1. the idea of extending from distillation to self-distillation is very natural and expected to work out well.

2. the idea of the asymmetry between how robust model and natural model influences each other is also very interesting, although the choices do not seem to be sufficiently discussed.

3. the model achieves reasonably good empirical performances.

**Weaknesses:**

1. the paper involves several interesting design that collaboratively contribute to strong performances, thus a more detailed ablation study (more than the one the authors offered) is probably necessary, for examples
    - why is the asymmetry of how robust model and natural model influence each other is necessary? The current ablation study only touches briefly on the removal of these losses. However, since the authors emphasized on this asymmetry, it will be quite essential to discuss what if we use the same way of how these models influences each other. For example, what if we use soft labels from natural model also, or what if when we train the natural models, we use KL regularization also.

2. The "LS" method does not seem to ever get spelled out, thus very hard to evaluate the relevant discussions. The paper cited is not immediately about the topics discussed in this paper.
    - As a result, I do not see how "Effects on utilizing last epoch predictions" in ablation study is relevant.

**Questions:**

1 the current ablation study in Figure 4 seems to suggest that self-distillation does not matter that much, a major performance boost comes from the natural model part, which seems quite counter-intuitive. It could be helpful if the authors explain more about this.

2. in table 1 and 2, it will be helpful to have a natural model for references.

---

> ### Author Response · Authors · 2023-11-16
> **Response to reviewer zap7**
>
> Thanks for your positive feedback and your suggestions about the experiments. We provide pointwise responses below.
>
> **The paper involves several interesting design that collaboratively contribute to strong performances, thus a more detailed ablation study (more than the one the authors offered) is probably necessary, for examples why is the asymmetry of how robust model and natural model influence each other is necessary? The current ablation study only touches briefly on the removal of these losses. However, since the authors emphasized on this asymmetry, it will be quite essential to discuss what if we use the same way of how these models influences each other. For example, what if we use soft labels from natural model also, or what if when we train the natural models, we use KL regularization also.**
>
>  We acknowledge that our ablation study may have been insufficient and, in response, have included two additional studies in Figure 4 in our revised version. Following your suggestion, we conducted a performance comparison experiment where the natural model in ROAD receives knowledge via KL-divergence similar to standard online distillation. Additionally, we reuse $\gamma$ as a hyper-parameter for the KL-divergence term to exchange symmetric knowledge between robust model and natural model. As shown in Table 1, regardless of the $\gamma$ value, our findings confirm that the original ROAD's asymmetric knowledge transfer method is superior to the symmetric knowledge exchange in natural accuracy.
>
> Table 1. Effect of transferring robust knowledge to natural model
> | Method               | NAT   | AA    |
> |----------------------|-------|-------|
> | ROAD ($\gamma = 1$)(KL) | 57.91 | 27.30 |
> | ROAD ($\gamma = 2$)(KL) | 60.00 | 27.65 |
> | ROAD ($\gamma = 3$)(KL) | 60.45 | 27.36 |
> | ROAD ($\gamma = 4$)(KL) | 60.65 | 27.36 |
> | ROAD                 | 62.09 | 27.60 |
>
> **The "LS" method does not seem to ever get spelled out, thus very hard to evaluate the relevant discussions. The paper cited is not immediately about the topics discussed in this paper. As a result, I do not see how "Effects on utilizing last epoch predictions" in ablation study is relevant.**
>
> “Effects on utilizing last epoch predictions” is designed to show the performance of self-adversarial distillation using last epoch predictions. We select label smoothing(LS) and adversarial knowledge distillation (AKD) as baselines because these methods also employ soft labels to improve robustness, similar to our method. However, we also agree that the corresponding ablation study does not need immediate attention, so we moved it to the appendix.
>
> **Figure 4 seems to suggest that self-distillation does not matter that much, a major performance boost comes from the natural model part, which seems quite counter intuitive.**
>
> While it may seem counterintuitive, it is actually because replacing the soft labels of the natural model with one-hot labels and allowing the natural model to unilaterally transfer knowledge to the robust model can actually lead to a detrimental decrease in robustness. However, it is important to note that utilizing self-distillation, as opposed to not using it at all, results in a performance increase of over 1\% in the perspective of PGD-20. We have updated Figure 4 to make this effect more readily apparent in our revised version.
>
> **It will be helpful to have a natural model for references.**
>
> Appreciate your suggestion! We add the performance of the natural model to Table 1 and Table 2 in our revised version.

---

### Official Review · Reviewer_cBNA · 2023-11-03

**Soundness:** 3 good
**Presentation:** 1 poor
**Contribution:** 3 good
**Rating:** 5
**Confidence:** 4

**Summary:**

This paper presents ROAD, an adversarial self-distillation approach designed to tackle over-confidence issues and improve the tradeoff between clean and robust accuracy of robust models in Adversarial Training (AT). The method moderates over-confidence by generating soft labels for adversarial examples, merging predictions from the last epoch model with the original hard labels. Additionally, to maintain clean accuracy, a regularizer is introduced that aligns the predictions of the robust model with those of a natural model on clean samples. Experimental results exhibit superior performance on both natural accuracy and robustness compared with both AT and Adversarial Distillation (AD) methods among various evaluated scenarios.

**Strengths:**

S1: This paper primarily focuses on adversarial self-distillation, a promising avenue for future adversarial machine learning research due to its lower resource demands compared to AD.

S2: This paper introduces a simple yet effective framework that excels in both natural accuracy and robustness for adversarial self-distillation. The exhibited experimental results also partially reveal the ineffectiveness of existing AD methods under this scenario.

**Weaknesses:**

W1: Certain key aspects of the presented theory in Section 3.1.1 appear ambiguous from my vantage point. The authors attempt to convince that the soft label proposed in Equation 1 can prevent the model’s predictions from becoming overly confident because the proposed soft label can implicitly provide smaller weights to adversarial samples which have a drastically increased confidence. The authors prove this by demonstrating the norm of the gradient of the loss w.r.t the output logits $ \frac{\partial \mathcal{L}}{\partial z_{t, i}^{\prime}} $ obtained with the proposed soft label will become smaller than that obtained with the original hard label if the confidence of the model output for $x\prime$ increases in this epoch. However, existing works generally propose to constrain a relatively larger norm of gradient $\frac{\partial \mathcal{L}}{\partial \theta_{t}} $ [1] to mitigate the overconfidence issue. It appears that the authors do not provide sufficient proof that the effect of the introduced gradient norm scaling factor w.r.t to the logit also works for the gradient w.r.t to the model weights $\theta$.

W2: Several statements presented in Section 3.2 are deemed to be either inaccurate or not adequately substantiated. For example, the referenced studies do not specifically support the claim that 'distilling knowledge from the static natural teacher may impair robustness.' It is strongly advised that the structure and argumentation of this section be thoroughly revised for clarity and accuracy.

W3: The logical progression within Section 3 is obscure, and the transitions between subsections lack fluidity, challenging the reader's ability to discern the author's objectives. Furthermore, the exposition accompanying Equation 3 falls short of providing sufficient elucidation or the underlying intuition for introducing the Robustness Enhancement term in the construction of the final objective function.

[1] Tao Li, Yingwen Wu, Sizhe Chen, Kun Fang, Xiaolin Huang: Subspace Adversarial Training. CVPR 2022: 13399-13408

**Questions:**

Q1: Considering Weakness W1, does the proof presented in Section 3.1.1 extend to ensuring that adversarial samples, which induce a significantly larger gradient norm with respect to the model weights $\theta$, can also be effectively penalized?

Q2: Because the paper appears to lack a comprehensive exploration of the tuning strategy of the hyperparameter $\lambda$ introduced in Equation 1, could you elucidate on the potential effects of employing a constant value for $\lambda$, or linearly increase the value of $\lambda$ instead of using the sine increasing schedule?

---

> ### Author Response · Authors · 2023-11-16
> **Response to reviewer cBNA**
>
> We appreciate about your detailed review and meaningful feedback. We provide pointwise responses below.
>
> **Considering Weakness W1, does the proof presented in Section 3.1.1 extend to ensuring that adversarial samples, which induce a significantly larger gradient norm with respect to the model weights $\theta$, can also be effectively penalized?**
>
> We have thoroughly read the suggested paper and appreciate its relevance to the broader field. However, we believe that directly linking our work to the findings of this paper may not be the most appropriate approach.
> The suggested paper addresses the issue of \textit{robust overfitting} which is a phenomenon that test accuracy on robustness drops significantly while train accuracy on robustness increases . In our understanding, the concepts of overconfidence and robust overfitting, while related, however are distinct. For instance, an overfitted model might exhibit low confidence when tested on unseen data from a different domain which is hard to say that the model is overconfident.
>
> Meanwhile, we have theoretically and empirically validated our assumption as detailed in our paper. To further support our argument, we have also included experiment results on the calibration performance of ROAD in the appendix D.3 in our updated paper.
>
> **Several statements presented in Section 3.2 are deemed to be either inaccurate or not adequately substantiated. For example, the referenced studies do not specifically support the claim that 'distilling knowledge from the static natural teacher may impair robustness.' It is strongly advised that the structure and argumentation of this section be thoroughly revised for clarity and accuracy.**
>
> Thanks for your detailed feedback! We have revised our sentences in Section 3.2. The sentences are
>
> ``However, experimental results presented in the literature [1,2] demonstrate that distilling knowledge from the static natural model can reduce robustness, indicating that it is not the proper approach."
>
> Also, we want to clarify that our statements in Section 3.2 are sufficiently substantiated. The referenced studies support the claim explicitly or implicitly that static natural teacher can reduce robustness. For example, In section 4.4 of [1], they provide experimental results that training with natural soft labels (NSLs) crafted by the natural model can significantly harm robustness of the student model and claim it in the last sentence in the paragraph. Additionally, In table 1 of [2], the table shows that the student model with a natural model teacher has lower accuracy under AutoAttack when compared with no teacher.
>
> We hope that the revised sentences address your concerns adequately.
>
> **The logical progression within Section 3 is obscure, and the transitions between subsections lack fluidity, challenging the reader's ability to discern the author's objectives. Furthermore, the exposition accompanying Equation 3 falls short of providing sufficient elucidation or the underlying intuition for introducing the Robustness Enhancement term in the construction of the final objective function.**
>
> We strongly agree that explanations about the method can be confusing to the readers. To enhance clarity and facilitate easier reading, we have added more details to the introductory part of Section 3, clearly delineating the roles of each subsection. In addition, we understand your concerns about Robustness Enhancement term and add proper elucidations. The sentences are
>
> ``This regularization term causes loss in natural accuracy as a trade-off for improved robustness. Nevertheless, this loss of accuracy can be recovered by the subsequent term."
>
> **Because the paper appears to lack a comprehensive exploration of the tuning strategy of the hyper-parameter
>  introduced in Equation 1, could you elucidate on the potential effects of employing a constant value for
> $\lambda$, or linearly increase the value of $\lambda$ instead of using the sine increasing schedule?**
>
> That is a nice feedback. Following your suggestion, we have conducted experiments that verify our selection of sine scheduling and updated in our paper. As shown in Table 1, utilizing last epoch predictions show an overall improvement in robustness
> compared to PGD-AT which trained with one-hot labels. In addition, we observe that the fixed policy results in over
> 1\% lower natural accuracy compared to other strategies.
>
> Table 1. Effect of scheduling interpolation ratio $\lambda$
> | Method        | NAT   | AA    |
> |---------------|-------|-------|
> | PGD-AT        | 57.23 | 25.13 |
> | Ours (Fixed)  | 56.20 | 26.16 |
> | Ours (Linear) | 57.55 | 25.88 |
> | Ours (Sine)   | 57.39 | 26.17 |
>
> Ref
>
> [1] Revisiting Adversarial Robustness Distillation: Robust Soft Labels Make Student Better, ICCV 2021
>
> [2] On the benefits of knowledge distillation for adversarial robustness, Arxiv 2022

---

### Author Response · Authors · 2023-11-21

We thank the reviewers for their constructive feedbacks. As the discussion period is near, please let us know if you have any further comments or concerns about our paper. We would be happy to answer your questions.

---

### Comment · Area_Chair_6zWC · 2023-11-21
**[Time Sensitive, ICLR24] Please read the authors' responses and try to discuss the remaining concerns with the authors**

Dear Reviewers,

The authors have provided detailed responses to your comments.

Could you have a look and try to discuss the remaining concerns with the authors? The reviewer-author discussion will end in two days.

We do hope the reviewer-author discussion can be effective in clarifying unnecessary misunderstandings between reviewers and the authors.

Best regards,

Your AC

---

### Meta-Review · Area_Chair_6zWC · 2023-12-06

**Metareview:**

While the paper addresses an important topic in the adversarial training domain and presents an interesting approach with adversarial self-distillation, it falls short in several critical aspects. The lack of novelty, insufficient detail in ablation studies, unclear methodology descriptions, incomplete comparative analysis, and omission of key benchmark evaluations significantly undermine the paper's impact and contribution to the field.

After the rebuttal, reviewers still do not feel that this paper contains enough novelty to the field. Given these substantial shortcomings, it is recommended that the paper be rejected. Future revisions should focus on addressing these concerns, particularly by enhancing the novelty of the approach, providing more comprehensive ablation studies, clarifying methodologies, and including thorough comparative analyses with both new and established methods.

**Justification For Why Not Higher Score:**

**Insufficient Novelty:** The paper overlaps significantly with prior works in areas such as the use of soft labels and online training methods. This lack of novelty detracts from the paper's contribution to the field.

**Justification For Why Not Lower Score:**

N/A

---

### Decision · Program_Chairs · 2024-01-16

Reject